behaviour/ecology

marine wildlife tourism, provisioning, additive models, behaviour, shark tourism

**Author for correspondence:**
Christine Legaspi
e-mail: t.legaspi@lamave.org

# In-water observations highlight the effects of provisioning on whale shark behaviour at the world's largest whale shark tourism destination

Christine Legaspi, Joni Miranda, Jessica Labaja, Sally Snow, Alessandro Ponzo and Gonzalo Araujo

Large Marine Vertebrates Research Institute Philippines, Cagulada Compound, Brgy. Tejero, Jagna, Bohol 6308, Philippines

CL, 0000-0002-2741-7282; GA, 0000-0002-4708-3638

The whale shark is the world's largest fish that forms predictable aggregations across its range, many of which support tourism industries. The largest non-captive provisioned whale shark destination globally is at Oslob, Philippines, where more than 500 000 tourists visit yearly. There, the sharks are provisioned daily, year-round, allowing the human–shark interaction in nearshore waters. We used in-water behavioural observations of whale sharks between 2015 and 2017 to understand the relationship between external stimuli and shark behaviour, whether frequency of visits at the site can act as a predictor of behaviour, and the tourist compliance to the code of conduct. Mixed effects models revealed that the number of previous visits at the site was a strong predictor of whale shark behaviour, and that provisioned sharks were less likely to exhibit avoidance. Compliance was poor, with 93% of surveys having people less than 2 m from the animal, highlighting overcrowding of whale sharks at Oslob. Given the behavioural implications to whale sharks highlighted here and the local community's reliance on the tourism industry, it is imperative to improve management strategies to increase tourist compliance and strive for sustainable tourism practices.

## 1. Introduction

The whale shark *Rhincodon typus* (Smith 1828) is the world's largest extant species of fish, inhabiting tropical and warm temperate

waters globally [1]. It is among the filter-feeding elasmobranchs whose broad diet includes copepods, fish eggs and crab larvae among others [1,2]. There have been records that the species also preys on anchovies and sardines, and other nektonic species like squid [2–5]. With their large bodies, they have evolved to improve foraging efficiency on these prey items [2]. They are also highly mobile, moving far and wide searching for food, as prey availability is patchy across their range [6]. They aggregate seasonally in areas linked to high food availability, despite their otherwise solitary nature [1,7]. Given their seasonal presence and their docile nature, whale sharks are an ideal candidate species for profitable wildlife tourism endeavours [1,8,9].

Whale shark watching is the second largest product of shark-based tourism, with millions of dollars annually brought to the global economy [9–11]. After the discovery of the seasonal presence of the species in Ningaloo Marine Park in the 1980s, where whale shark tourism was first established [12], the whale shark tourism industry has since grown, making substantial contributions to economies as seen in countries like Australia [10], Maldives [11], Mexico [13], Philippines [14,15] and Seychelles [16]. Aside from this, the industry also provides an avenue for scientific research and conservation outcomes [12,14–18].

Despite the economic advantages that the industry provides, direct effects of tourism on the host species have been reported by the majority of whale shark behavioural studies within tourist hotspots [14,17,19–24]. Whale sharks react (through banking, diving or changing direction) in the presence of tourists in Ningaloo, Australia [19]. Similarly, the whale sharks in Donsol, Philippines are likely to be disrupted from feeding, displaying avoidance behaviour during tourist interactions [14]. Whale sharks in Mozambique and Southern Leyte, Philippines, displayed avoidance when tourist vessels were within close proximity [17,24], and during the majority of tourist encounters, suggesting a change in the behaviour of the sharks on at least a short temporal scale [20,24]. These observations, however, are contrary to the behaviours observed in provisioned whale sharks in Oslob, Philippines [21,22].

Provisioning, or feeding, wildlife for tourism allows people to experience close and predictable encounters with otherwise elusive species [25]. Oslob, Philippines is currently the world's largest non-captive provisioned whale shark tourism destination, receiving over 500 000 tourists in 2018 with an estimated US$10 million in ticket sales [18,26]. This community-run site is active year-round, and although some tourist [26] and whale shark [23] seasonality is observed, the sharks are provisioned daily. While the activity offers an avenue for a tourism industry to flourish [26], there is evidence that the current tourism practices have impacts on the host species [27–29]. Behaviour modification is evident through significant differences in residency patterns between provisioned and non-provisioned individuals [21], with some individuals displaying year-round residency, while others are seasonal residents [23]. Provisioned individuals have also increased tolerance to human interactions and are less likely to display avoidance behaviour [22]. High presence of scars, and changes in diving behaviour and metabolic rate have also been reported in provisioned individuals [23,27,28]. Furthermore, poor tourist compliance, especially regarding minimum distance, is evident, which suggests high tourism pressure on the whale sharks [22]. These considerable impacts [21–23,27,28] on an endangered and nationally protected species should prompt stakeholders, such as the tour operators and the local government, to delineate limits of acceptable change or other management strategies.

In this study, we used in-water behavioural observations of whale sharks through dedicated focal follows [30] at Oslob, Cebu, Philippines from February 2015 to May 2017 to further understand the relationship between tourism activities and animal behaviour, and tourists' compliance to the code of conduct currently in place at the site that is dictated by a local ordinance. We further investigated whether shark behaviour is also affected by environmental and anthropogenic variables.

# 2. Methods

## 2.1. Study site

The local community has managed the whale shark tourism in Oslob since 2011 [15]. The whale shark interaction area is situated in *barangay* (village) Tan-awan, Oslob, Cebu, Philippines (figure 1). Tourist visitation rapidly increased from 98 000 in 2012 to 508 000 in 2018 [26], with tourist numbers fluctuating seasonally corresponding with national and international holiday periods (electronic supplementary material, figure S1). The whale sharks in Tan-awan are provisioned daily, year-round, between 6.00 and 12.00 within a demarked interaction area [21]. Between 250 and 400 kg of feed composed mainly of sergestid shrimp species—locally known as *uyap* or *uyabang*—is used to attract whale sharks and

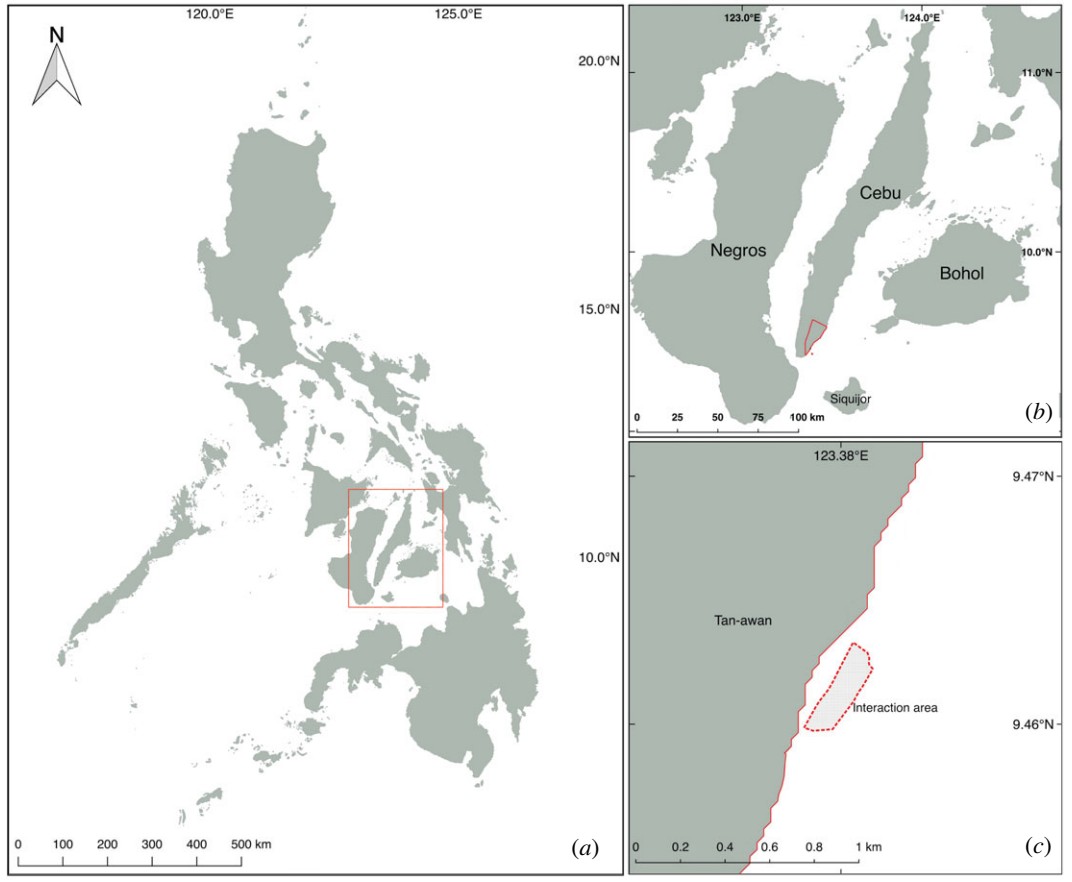

**Figure 1.** Map of the study site. Map of the Philippines (*a*), the island of Cebu (*b*), Oslob and *barangay* Tan-awan (*c*), and the interaction area (demarked site with a dashed line).

facilitate close encounters with the species. The overall duration of the feeding time depends on the number of sharks and tourists present during the day. Tourists who wish to snorkel or watch from a boat are briefed regarding the interaction guidelines before they are queued and assigned to a particular boat with two to three guides for the whale shark activity [22].

## 2.2. Code of conduct and compliance

The whale shark watching operations in Tan-awan, Oslob are governed through a municipal ordinance (Ordinance No. 091 of 2012, Local Government Unit, Oslob), where a code of conduct for in-water whale shark interaction is provided. The code of conduct states that the duration of the activity, for both swimmers and watchers, is limited to 30 min. However, no such time limit is set for SCUBA divers. Only non-motorized vessels are allowed within the demarked interaction area at any time, but no regulation is in place for motorized vessels carrying the tourists to and from the interaction area boundary. The ordinance includes the minimum distance from the shark during interaction—minimum of 5 m from the side and tail of the animal, and a minimum of 2 m from the mouth. The maximum number of tourists allowed to interact with one whale shark is six, while four if SCUBA diving. The local ordinance, however, fails to state a distance within which the maximum number of tourists must be. Given the 5 m minimum distance rule, 10 m was chosen as the maximum distance to determine compliance to the maximum number of tourists per shark to feasibly count swimmers, divers and boat holders. This is a conservative measure as in-water visibility varies throughout the year. Furthermore, the ordinance also states that there should be no touching or riding of the whale sharks, no splashing, and only feeders are allowed to feed the whale sharks.

## 2.3. Focal follows

Following methods adapted from a previous study at the site [22], researchers conducted focal follow surveys for 20 min to record the behaviour of whale sharks within the interaction area, and the

**Table 1.** Events and behaviours recorded in each focal follow [21].

| code | predominant behaviour | definition |
|------|----------------------|------------|
| HF | horizontal feeding | Shark actively swims behind feeder boat with its body angled horizontally (variation of angle depending on speed of current and feeder boat positioning). |
| VF | vertical feeding | Shark is in a stationary position, with its body in a vertical orientation with its mouth just below the water surface. Food is ingested by gulping water using a suction technique. |
| NF | natural feeding | Shark swims with either partially or totally open mouth displaying passive or active feeding in an area away from the feeder boats. |
| FS | free swimming | Shark swims with mouth closed, independently of feeder boat proximity or food availability in the water. |
| *events* | | |
| AT | active touch | Guest intentionally approaches the animal and initiates shark contact with any body part or gear (i.e. fins, camera, camera pole). |
| PT | passive touch | Any contact between shark and guest where the guest does not intentionally approach the animal. |
| FC | feeder contact | Feeder intentionally touches the shark with any body part or gear different to the feeder. |
| S2S | shark-to-shark contact | Two or more sharks make physical contact. |
| BC | boat contact | Physical contact between shark and any boat in the interaction area. |
| RB | roadblock | Guest or boatman blocks the natural path of shark. |
| *reactions* | | |
| NR | no reaction | No evident behavioural change recorded immediately after the observed event |
| SW | swam off | Shark changes behaviour abruptly and swims away without depth variation |
| DV | dive | Shark changes behaviour and dives |
| BK | bank | Shark rolls and orientates its dorsal side towards the perceived threat |
| CG | cough | Shark forcefully expels water and other material out of the mouth |
| ER | eye roll | Shark rolls eye backward into the socket |
| VS | violent shudder | Shark physically shakes its body |

compliance of tourists during the tourism activity. This was conducted between February 2015 and May 2017. Effort was not consistent and was dependent on researcher availability, but weekend (busy) and weekdays were targeted equally to reflect potential differences. Within each survey, the assigned researcher followed the first shark randomly encountered within the interaction area and identified through photo-identification (photo-ID) [21]. Following photo-ID, researchers would then observe and record the shark's predominant behaviour and any events that occurred, followed by the shark's reaction to each event (table 1). Each reaction was considered as the sharks' display of avoidance behaviour, while no reaction was considered otherwise. Tourist compliance with the local code of conduct was recorded every 5 min within each 20 min survey, and the number of tourists present within 2 m anywhere around the shark (overall), within 2 m in front of the shark, within 5 m from side to tail and within 10 m overall. The tourists were categorized as either boat holders, swimmers, or divers.

## 2.4. Data analyses

We fitted a binomial generalized linear mixed model (GLMM) using the package *lme4* in R [31,32], to investigate variables influencing whale shark behaviour. Whale shark predominant behaviour (table 1)

was coded as either vertical and horizontal feeding from a feeder boat (1) or free swimming or naturally feeding in the interaction area (0). Potential exploratory variables included previous visits at the site (the number of days that an individual was identified at the study site, as confirmed through photo-ID); current (1 if there is no current, 2 if there is effort needed to swim, 3 if there is difficulty in swimming, 4 if there is a need to hold on to boats, 5 if it is impossible to swim at all); visibility (m); sea surface temperature (SST) (using weekly averages of 0.5 latitudinal degree by 0.5 longitudinal degree from NOAA's Optimum Interpolation SST v. 2, NOAA/OAR/ESRL PSD, Boulder, CO, USA, https://www.esrl.noaa.gov/psd/); sex (M or F); estimated size (m); and presence of major scars (1 or 0) [27]. The shark ID was added as a random effect to avoid pseudoreplication [33]. We used the *drop1* function ($\chi^2$) for variable selection tested at $p < 0.05$, and ANOVA tests for model selection using the Akaike information criterion (AIC) [34]. SST served as a seasonality variable in the models.

We then fitted a second binomial GLMM to understand which factors (table 1) affect whale shark response (presence or absence of avoidance behaviour) to stimuli. Exploratory variables (table 1) included event; predominant shark behaviour; previous visits (days) at the site since 31 March 2012, when daily monitoring surveys began; number of swimmers less than 10 m of the shark; number of divers less than 10 m of the shark and number of boat holders less than 10 m from the shark. Shark ID was used as a random effect in the model. Variable and model selections were the same as described above.

# 3. Results

From the 358 twenty-minute focal follow surveys completed, 46 individual whale sharks, as determined through photo-ID, with a mean total length of 4.1 m (± 1.1 s.d., range 2.5–6.0 m), were observed. Most whale sharks were male (80%) highlighting male bias in the data collected ($\chi^2 = 71.82$, $p < 0.001$) reflecting the nature of this aggregation [21]. Previous visits at Tan-awan on the day of focal follow ranged from 1 to 1321 (median 336 days, 328 s.d.). On the days where focal follow surveys were conducted, there was a mean of $14 \pm 5.49$ individuals (range 5–31 individuals) present in the study area.

## 3.1. Predictors of behaviour

From a total of 358 complete focal follows, whale sharks were predominantly observed horizontal feeding (71%), vertical feeding (14%) and free swimming (15%). The GLMM indicated that the number of previous visits was a significant predictor that whale sharks would be observed vertical or horizontal feeding from a feeder boat ($p < 0.01$; figure 2). The model also indicated SST as a predictor at $p < 0.1$ (table 2).

## 3.2. Avoidance behaviour

We recorded a total of 692 events throughout the study period: 38 active touches, 301 passive touches, 125 boat contacts, 55 feeder contacts, 30 roadblocks and 143 shark-to-shark contacts. Whale sharks were more likely to exhibit an avoidance behaviour (dive, swam off, eye roll, violent shudder) in response to a shark-to-shark contact ($2.03 \pm 0.53$ s.e., $p < 0.01$) or a roadblock to their path ($2.01 \pm 0.64$ s.e., $p < 0.01$; table 2; electronic supplementary material, figure S2). Contrastingly, whale sharks were less likely to display an avoidance behaviour if they were vertical ($-2.24 \pm 0.38$ s.e., $p < 0.01$) or horizontal feeding ($-1.98 \pm 0.31$ s.e., $p < 0.01$) from a feeder boat (table 3). Other variables were not significant.

## 3.3. Compliance

Compliance to the regulations set out by the local government of Oslob (Ordinance No. 091 of 2012), was recorded for all 358 focal follows. At least one swimmer was observed within 2 m of the shark on 75.1% of surveys, while at least one diver was observed within 2 m of the shark on 13.4% of surveys. Boat holders were observed within 2 m of the shark on 84.9% of surveys. There was an average of 5.4 swimmers (range 0–25), 2.0 divers (range 0–17) and 9.9 boat holders (range 0–41) within 10 m (overall) from the shark. There was a mean of 17.3 tourists (range 2–55) observed to be within 10 m from one shark, the maximum number of people per shark according to the ordinance is six (figure 3). Data, in comparison with previous years, are summarized in table 4.

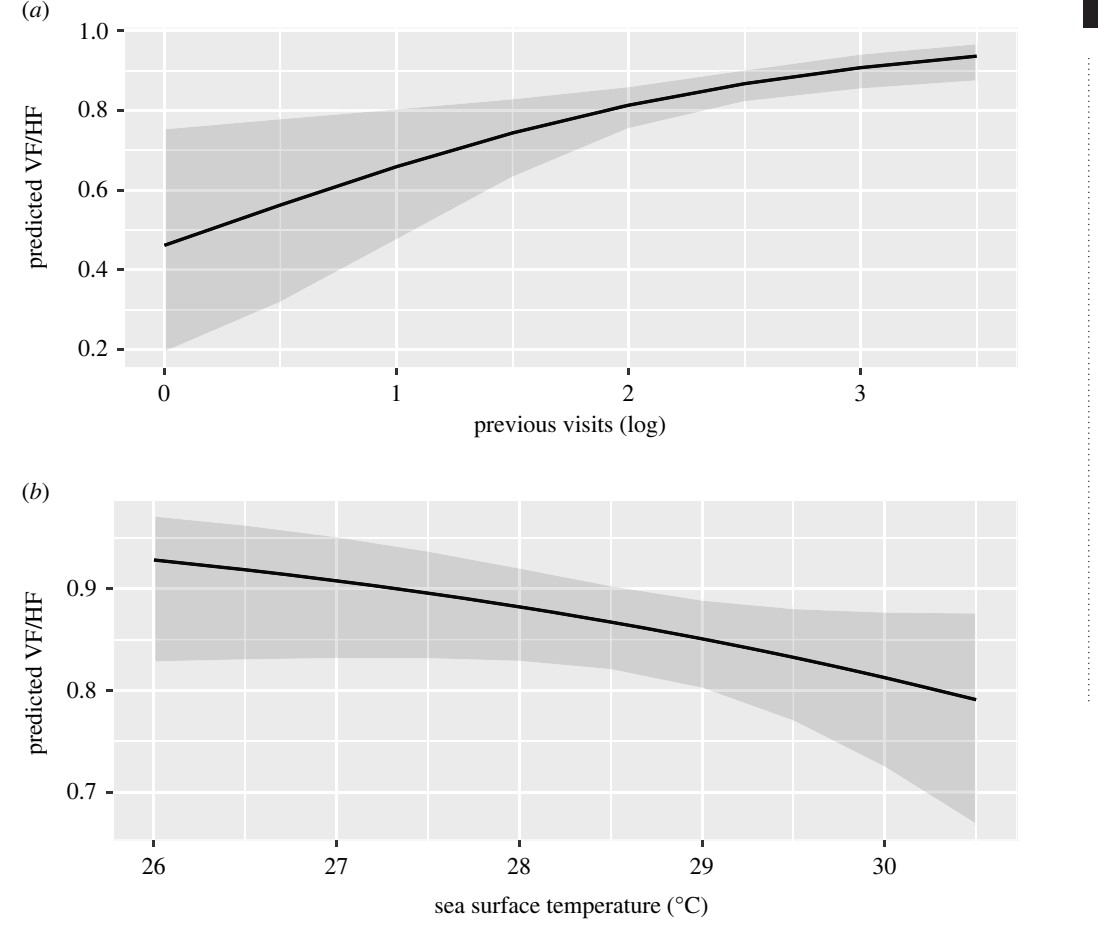

**Figure 2.** Predicted values for vertical and horizontal feeding behaviour from GLMM against previous visits (*a*) and sea surface temperature (*b*).

**Table 2.** Significant parameter estimates of the GLMM for predicting whale shark behaviour (± s.e.). Statistical significance of *p*-values is indicated by asterisks.

| fixed effect | coefficient (± s.e.) | *p*-value |
|---|---|---|
| *Intercept* | 7.69 (4.78) | 0.11 |
| previous visits (days) | 0.81 (0.28) | <0.01* |
| sea surface temperature (°C) | −0.27 (0.17) | 0.09** |

*\*p < 0.01, \*\*p < 0.1.*

## 4. Discussion

Our results show the impacts of provisioning on the behaviour of whale sharks in Oslob, and the effects of the tourism industry failing to comply with the regulations. These results are strong indicators that management intervention is necessary. Similar to many other whale shark hotspots globally, the whale shark population in Oslob is predominantly composed of juvenile males [1,21], hence an observed male bias during the focal follows. Unlike many other sites, however, whale sharks here are fed daily, year-round, between 6.00 and 12.00, which is the reason for their prolonged residency periods [21,23]. Interestingly, this study revealed that the sharks' number of previous visits at the site is a significant predictor of the sharks' feeding behaviour from the boats, complementing the results from a previous study where there were significant observations of vertical feeding in resident sharks in Oslob [22]. This complements observations by researchers wherein newly identified whale sharks were mostly seen free swimming and not feeding. Oslob whale sharks with long visitation history

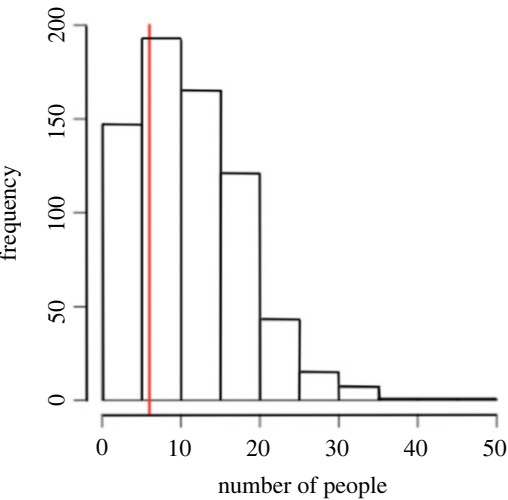

**Figure 3.** Frequency distribution of the number of tourists seen within 10 m from the sharks. The red line shows the recommended maximum number of people per shark (six).

**Table 3.** Parameter estimates of the GLMM for avoidance behaviour (± s.e.). Statistical significance of *p*-value is indicated by asterisks.

| fixed effect | coefficient (± s.e.) | *p*-value |
| --- | --- | --- |
| *Intercept* | 0.15 (0.74) | 0.84 |
| event: boat contact | 0.08 (0.56) | 0.88 |
| event: feeder contact | 0.03 (0.63) | 0.96 |
| event: passive touch | −0.12 (0.52) | 0.81 |
| event: roadblock | 2.01 (0.64) | <0.01* |
| event: shark-to-shark contact | 2.03 (0.53) | <0.01* |
| predominant behaviour: horizontal feeding | −1.98 (0.31) | <0.01* |
| predominant behaviour: vertical feeding | −2.24 (0.38) | <0.01* |
| previous visits (days) | −0.01 (0.11) | 0.93 |
| no. of boat holders < 10 m | −0.02 (0.02) | 0.28 |
| no. of swimmers < 10 m | −0.02 (0.03) | 0.44 |
| no. of divers < 10 m | −0.05 (0.07) | 0.52 |

*$p < 0.01$.

**Table 4.** Level of non-compliance observed in Oslob whale shark watching focal follows between February 2015 and May 2017, with some comparison with previous years where available [21].

| regulation | % non-compliant | | | | mean (s.d., range) |
| --- | --- | --- | --- | --- | --- |
| | *this study* | *2014* | *2013* | *2012* | *this study* |
| swimmer < 2 m from shark | 75.1 | 84.9 | n.a. | 65.6 | 2.6 (2.7, 0–18) |
| diver < 2 m from shark | 13.4 | 20.2 | n.a. | 26.3 | 0.4 (1.2, 0–12) |
| boat holders < 2 m from shark | 84.9 | 76.5 | n.a. | 54.3 | 4.2 (3.1, 0–14) |
| *swimmer + boat holder < 2 m from shark* | 92.7 | 96.6 | 90.7 | 78.6 | 4.3 (3.8, 0–22) |

were seen present at the interaction site even before the day's provisioning activities begin, displaying anticipatory behaviour as they associate the site with food [22]. Whale sharks, in general, are known to display different feeding techniques depending on the prey abundance and composition [2]. Since whale sharks are opportunistic foragers and are adaptable to prey availability and abundance [7], we

conclude from our results that whale sharks frequenting Oslob are able to modify their behaviour in response to the food being provisioned for tourism.

Our results also highlight that whale sharks are more likely to display avoidance in the event of a roadblock, wherein the animal's direction of travel is obstructed. This is similar to observations in Donsol, a non-provisioned seasonal aggregation in the Philippines, where path obstruction was one of the main causes of avoidance behaviour [14]. Contrastingly, the individuals that were observed feeding (either vertically or horizontally) were less likely to display avoidance behaviours and react to external stimuli. This corroborates a previous study in Oslob where whale sharks learnt to associate food with the site and were less likely to display avoidance behaviour as a response to the provisioning linked at the site [22]. The whale sharks' learning abilities have yet to be further examined, although there have been records of other elasmobranch species which display different learning techniques. Small-spotted cat sharks (*Scyliorhinus canicula*) have shown their learning capability to improve foraging opportunities through the presence and absence of positive reinforcement [35]. Blacktip reef sharks (*Carcharhinus melanopterus*) learn to survive against fishing pressures through experience in terms of their catchability [36]. Juvenile Port Jackson sharks (*Heterodontus portusjacksoni*) improve foraging success where they discover new foraging tracks through social learning, despite them being solitary as a species [37]. Overall, sharks are capable of learning to improve food search and remember spatio-temporal information for survival [38], suggesting the whale sharks in Oslob learn and modify their behaviour to exploit a new foraging opportunity at the tourist site.

Whale sharks move far and wide, with a recent study highlighting their international movements between the Philippines and Malaysia [5,39]. The seasonal movement and feeding plasticity of whale sharks, that also involves behavioural shifts, suggest the whale sharks are highly capable of adapting to new food sources available [7,39]. Ephemeral pulses of primary productivity are patchily distributed in the tropics and thus, the ability to adapt behaviour to capitalize on a foraging opportunity is essential [7,22]. The low productivity during *habagat,* Southwest Monsoon (June–September), also coincides with the high influx of whale sharks in Oslob [23]. During these months, there is higher SST in the area (approx. 3–4°C higher than during the *amihan,* Northeast Monsoon, November–May). This would explain our results that SST is a predictor of the whale sharks' feeding behaviour. Since there are more whale sharks in the interaction area during these months, the competition for feeding opportunities from the provisioning boats is higher, resulting in an overall reduction in the vertical or horizontal feeding behaviour observed. When in Oslob, the whale sharks generally spend the majority of their time at the surface because that is where the food is being provisioned [28]. Whale sharks have a need to thermoregulate as ectotherms [40], and the prolonged exposure to warmer waters [28], particularly during *habagat* when the water is warmer, might also explain their reduced predictability feeding from the food provisioned. The predictability of whale shark feeding behaviour in Oslob is therefore a reflection of the consistent provisioning of food, and as such, offers unique insights into their versatile nature.

Provisioning for tourism facilitates and allows encounters with wild animals, such as whale sharks, that appeals to tourists as they are able to encounter them closely at their convenience [15,25,26]. With 100% guaranteed daily presence of whale sharks, Oslob has entertained more than 500 000 tourist visits in 2018 [26] with an average visitation of 1415 ± 454 (range 301–3024) tourists per day. This has been the highest recorded number yet in terms of tourist visits, making the municipality the largest non-captive provisioned whale shark watching destination in the world. Overcrowding, as a consequence of the high volume of tourist visitation, was highlighted in a previous study where 95% of the tourists who visited Oslob perceived that the interaction area was overcrowded [26]. Our results showed an average of 17.3 tourists were within 10 m from a shark—almost twice the recommended number in the local ordinance (a limit of six snorkelers and four divers per shark). Even though the presence of tourists within 10 m was not a significant predictor for whale shark avoidance, it would add the probability of roadblocks, a form of disturbance, to which the whale sharks significantly responded to. Furthermore, it is important to note that since that data was collected, the site has doubled in terms of numbers of tourists, suggesting a larger overcrowding issue at the time of writing. In terms of non-compliance to distance-to-shark of swimmers and boat holders, our results show that non-compliance to distance remains high at 92.7%. The code of conduct functions as a mitigation strategy to lessen tourism impacts on whale sharks [14]. To date, there has been a neglect on minimizing tourism impacts on the endangered whale shark through a lack of enforcement of the code of conduct currently governed by a local ordinance.

This study reveals the effects of provisioning and the tourism activities on the behaviour of whale sharks and the poor compliance of tourists to the code of conduct in Oslob, Philippines. Results show the sharks' ability to associate the food source with the site, through provisioning. Individuals with longer previous visits were predictably observed feeding, noting that feeding is a modified behaviour in

response to provisioning as individuals with less previous visits at the site were less likely to be observed feeding from the provisioning boats. This is in addition to their anticipatory presence [22]. There is also evidence of their modified behaviour in response to the mass-tourism site, as feeding individuals are less likely to display avoidance behaviour to maximize food reward while being provisioned. Furthermore, Oslob is perceived as overcrowded, indicating a high level of tourism pressure towards the sharks being provisioned [18,22]. The changes to the natural behaviour of an endangered species through tourism, and the poor tourist compliance, highlight a need for a new management approach.

# 5. Tourism management implications

While it is important to note that the tourism brought economic benefit to the local community, the level of impact from the industry on the whale sharks should not be overlooked [15]. Updating and improving tourism guidelines, and how they can be enforced, should be taken into consideration. Given that roadblocks and shark-to-shark contacts cause the most disturbances to whale sharks at Oslob, the operators should look into effectively enforcing all tourists to hold onto the boats or to stay in a designated area to prevent overcrowding and obstruction of the sharks' movements. Oslob attracts more than 500 000 tourists per year, where the whale sharks and the marine environment are used as a source of revenue from this tourism industry. It should be in all stakeholders' interest to ensure best practices are followed based on best available science, experts' recommendations and the local community's needs. A compromise by all stakeholders is necessary to improve the tourism in Oslob and reduce the impacts on the host species.

The Philippines is also a signatory country to the Convention on Migratory Species (CMS). The country has aligned to the convention's documents including the 'Concerted Action for the whale shark (*Rhincodon typus*)' [41], where provisioning of whale sharks is highlighted as an unsustainable practice for wildlife tourism that 'needs to be regulated either through prohibitions or limiting/minimizing these activities'. In addition, a joint memorandum circular regarding marine wildlife tourism conduct has recently been passed by different national departments, namely the Department of Tourism, the Department of Environment and Natural Resources, the Department of Interior and Local Governance, and the Department of Agriculture. It specifies the best practices for managing marine wildlife tourism, including the whale shark (DOT-DA-DILG-DENR Joint Memorandum Circular No. 01 series of 2020). It should thus be a priority for National and Regional governments to properly and effectively regulate tourism with the whale shark, especially in Oslob that receives record tourist numbers, and the aforementioned Joint Memorandum Circular delineates the way to do so.

Ethics. This study was conducted with methods in compliance with national and municipal legislation. The methods employed herein were minimally invasive in nature and no animal was restrained. The work was done following guidelines, and in collaboration with the Department of Agriculture–Bureau of Fisheries and Aquatic Resources, and with the prior informed consent of the local government unit of Oslob.

Data accessibility. Whale shark identification data is openly available on 'Wildbook for Whale Sharks', an open-source database on www.whaleshark.org. Our data are also deposited at Dryad Digital Repository: https://dx.doi.org/10.5061/dryad.fn2z34tqs [42].

Competing interests. We declare we have no competing interests.

Funding. We received no funding for this study.

Acknowledgements. This study was done in collaboration with the local government unit of Oslob represented by the honourable mayor Jose C. Tumulak Jr, the TOSWFA boatmen association, the Department of Agriculture–Bureau of Fisheries and Aquatic Resources, and the Department of Environment and Natural Resources. We thank the staff and volunteers of Large Marine Vertebrates Research Institute Philippines, without whom none of this work would have been possible.

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
