## [Reviewer comments · Royal Society Open Science]

Review History

RSOS-200392.R0 (Original submission)

Review form: Reviewer 1 (Olivia Lee)

Is the manuscript scientifically sound in its present form?

Yes

Are the interpretations and conclusions justified by the results?

Yes

Is the language acceptable?

Yes

Do you have any ethical concerns with this paper?

No

Have you any concerns about statistical analyses in this paper?

No

Recommendation?

Major revision is needed (please make suggestions in comments)

Comments to the Author(s)

I think this manuscript provides timely and important information on species behavior and management recommendations for managing the whale shark tourism industry. I think the paper would benefit from a little more introductory explanation about the seasonality of tourism pressure (i.e. is there a low and high tourist season? What is the magnitude of tourism pressure between seasons/ between locations in the Philippines?) I also think the manuscript could use a little more explanation in the data analysis to improve clarity but to also support future meta-analyses of similar studies in the future, and more critical editing of the discussion section to improve clarity in the content (see Appendix A).

Review form: Reviewer 2 (Simon Pierce)**Is the manuscript scientifically sound in its present form?**

Yes

Are the interpretations and conclusions justified by the results?

No

Is the language acceptable?

Yes

Do you have any ethical concerns with this paper?

No

Have you any concerns about statistical analyses in this paper?

No

Recommendation?

Accept with minor revision (please list in comments)

Comments to the Author(s)

General comments:

I very much enjoyed reading this study on what has become quite a controversial topic in marine wildlife tourism: provisioning whale sharks. The scientific content has been well implemented. However, I think the discussion, as it is now, reads as more opinionated than the results – taken in isolation – warrant. That said, I am not unsympathetic to the authors' opinion, and I have a couple of general comments that may be useful to consider.

First, my read of this study – again, taken in isolation – is that the primary takeaway is that 'habituated whale sharks show minimal avoidance behaviours'. In itself, this doesn't appear to show a clear negative impact to the sharks. In the introduction (L76-78) the authors contend that local stakeholders should delineate limits of acceptable change – this would be a useful point to expand on within the discussion, in light of the results of this study. A more explicit discussion of real or potential (i.e. based on the broader literature) impacts would also be helpful. This is mentioned in the introduction, but could be partly revisited in the discussion based on these new results.

Second, there are clear, systematic breaches of the code of conduct here. In terms of recommendations, I think there is a stark choice for managers – either (a) acknowledge that these ‘rules’ are viewed as unnecessarily precautionary in practice, or (b) take steps to provide an equally systematic solution. To me, the results presented here suggest that a useful measure might be to have all visitors hold onto the boats, or a rope between boats, while feeders slowly move up and down the resulting controlled line of people. Fig 4 shows that RB and S2S cause the most disturbance. Controlling the distance, which appears to be within the feeders control (i.e. L230-232), would reduce most of the ‘events’ listed, while potentially minimising crowding perceptions too, if all the tourists had an unobstructed view of the sharks. That would space out the sharks from one another, and from the people, rather than the free-for-all demonstrated in this study. This is just a suggestion, but, in that spirit, I think more tangible and practical recommendations in the discussion would benefit all parties.

I would also change the MS title to more directly address these points.

I hope this helps.

Simon Pierce.

Specific comments:

Abstract:

L15-16 - Relationship *between*. Seasonality rather than periodicity?

L20-21 - I'd say ‘overcrowding’ rather than ‘high tourism pressure’, as it's unclear from these results whether this is actually affecting the sharks, but overcrowding is clear based on the site's own rules.

L23-24 - It doesn't necessarily hold that this is unsustainable, depending on how that term is being defined here. The sharks keep coming back, for one thing, and there isn't necessarily a decline in tourist numbers. You could reword this conclusion to better reflect the direct results of this study.

Introduction:

L34-35 - This makes their size sound like a bug, rather than a feature – whale sharks are likely to be large to enhance their foraging efficiency.

L41 - ‘Product’ rather than ‘market’?

L46 - Australia is effectively featured twice in this sentence.

L50 - Effects rather than implications.

4th Para - How does this study differ and build on previous work, such as refs 21 and 22? Given the amount of work that has taken place at this site, a more detailed introduction would be useful for placing the current study in context.

Methods:

Table 1 - The new study by Tomita et al. (2020, PLOS ONE) describes this (ER); it's the whole eye, not the pupil.

Results:

Fig 2 - I don't understand this? Frequency? There are only 46 sharks. Is this a bar for each individual shark? Seems like a different type of graph should be used here.

Discussion:

The first two sentences are repetitive from the introduction. Better to lead with the main results.

L273 - Rewording could help make this point clearer.

L303-304 - I'm not really making the connection between SST as a prediction of occurrence with this particular point. You're seeing more sharks in warmer water; isn't this arguing the opposite, if you're saying the sharks need to cool down following a long exposure to warm-water in the provisioning area?

L329 - Is the Quiros ref relevant here? As used here, it seems to state the COC like it works, whereas this reference examined adherence.

L333 - I'd argue that it's testing whale shark responses to tourists etc, rather than revealing implied negative impacts.

L343-345 (and onwards) - Or, potentially, a *changed* management approach, as I've noted in my general comments.

Decision letter (RSOS-200392.R0)

Dear Ms Legaspi

The Editors assigned to your paper RSOS-200392 "In-water observations highlight the effects of provisioning on whale shark behaviour at the world's largest whale shark tourism destination" have now received comments from reviewers and would like you to revise the paper in accordance with the reviewer comments and any comments from the Editors. Please note this decision does not guarantee eventual acceptance.

Please submit your revised manuscript and required files (see below) no later than 21 days from today's (ie 10-Aug-2020) date. Note: the ScholarOne system will 'lock' if submission of the revision is attempted 21 or more days after the deadline. If you do not think you will be able to meet this deadline please contact the editorial office immediately.

on behalf of Dr Asha de Vos (Associate Editor) and Pete Smith (Subject Editor)
openscience@royalsociety.org

Associate Editor Comments to Author (Dr Asha de Vos):

Associate Editor: 1

Comments to the Author:

This paper looks at a topic of great interest to many. The reviewers have taken time to provide constructive criticism that I believe will benefit the paper. Please do go ahead with the suggested revisions. Thank you

Reviewer comments to Author:

Reviewer: 1

Comments to the Author(s)

I think this manuscript provides timely and important information on species behavior and management recommendations for managing the whale shark tourism industry. I think the paper would benefit from a little more introductory explanation about the seasonality of tourism pressure (i.e. is there a low and high tourist season? What is the magnitude of tourism pressure between seasons/ between locations in the Philippines?) I also think the manuscript could use a little more explanation in the data analysis to improve clarity but to also support future meta-analyses of similar studies in the future, and more critical editing of the discussion section to improve clarity in the content.

Reviewer: 2

Comments to the Author(s)

General comments:

I very much enjoyed reading this study on what has become quite a controversial topic in marine wildlife tourism: provisioning whale sharks. The scientific content has been well implemented. However, I think the discussion, as it is now, reads as more opinionated than the results – taken in isolation – warrant. That said, I am not unsympathetic to the authors' opinion, and I have a couple of general comments that may be useful to consider.

First, my read of this study – again, taken in isolation – is that the primary takeaway is that 'habituated whale sharks show minimal avoidance behaviours'. In itself, this doesn't appear to show a clear negative impact to the sharks. In the introduction (L76-78) the authors contend that local stakeholders should delineate limits of acceptable change – this would be a useful point to expand on within the discussion, in light of the results of this study. A more explicit discussion of

real or potential (i.e. based on the broader literature) impacts would also be helpful. This is mentioned in the introduction, but could be partly revisited in the discussion based on these new results.

Second, there are clear, systematic breaches of the code of conduct here. In terms of recommendations, I think there is a stark choice for managers – either (a) acknowledge that these ‘rules’ are viewed as unnecessarily precautionary in practice, or (b) take steps to provide an equally systematic solution. To me, the results presented here suggest that a useful measure might be to have all visitors hold onto the boats, or a rope between boats, while feeders slowly move up and down the resulting controlled line of people. Fig 4 shows that RB and S2S cause the most disturbance. Controlling the distance, which appears to be within the feeders control (i.e. L230-232), would reduce most of the ‘events’ listed, while potentially minimising crowding perceptions too, if all the tourists had an unobstructed view of the sharks. That would space out the sharks from one another, and from the people, rather than the free-for-all demonstrated in this study. This is just a suggestion, but, in that spirit, I think more tangible and practical recommendations in the discussion would benefit all parties.

I would also change the MS title to more directly address these points.

I hope this helps.

Simon Pierce.

Specific comments:

Abstract:

L15-16 - Relationship *between*. Seasonality rather than periodicity?

L20-21 - I'd say ‘overcrowding’ rather than ‘high tourism pressure’, as it's unclear from these results whether this is actually affecting the sharks, but overcrowding is clear based on the site's own rules.

L23-24 - It doesn't necessarily hold that this is unsustainable, depending on how that term is being defined here. The sharks keep coming back, for one thing, and there isn't necessarily a decline in tourist numbers. You could reword this conclusion to better reflect the direct results of this study.

Introduction:

L34-35 - This makes their size sound like a bug, rather than a feature – whale sharks are likely to be large to enhance their foraging efficiency.

L41 - ‘Product’ rather than ‘market’?

L46 - Australia is effectively featured twice in this sentence.

L50 - Effects rather than implications.

4th Para - How does this study differ and build on previous work, such as refs 21 and 22? Given the amount of work that has taken place at this site, a more detailed introduction would be useful for placing the current study in context.

Methods:

Table 1 - The new study by Tomita et al. (2020, PLOS ONE) describes this (ER); it's the whole eye, not the pupil.

Results:

Fig 2 - I don't understand this? Frequency? There are only 46 sharks. Is this a bar for each individual shark? Seems like a different type of graph should be used here.

Discussion:

The first two sentences are repetitive from the introduction. Better to lead with the main results.

L273 - Rewording could help make this point clearer.

L303-304 - I'm not really making the connection between SST as a prediction of occurrence with this particular point. You're seeing more sharks in warmer water; isn't this arguing the opposite, if you're saying the sharks need to cool down following a long exposure to warm-water in the provisioning area?

L329 - Is the Quiros ref relevant here? As used here, it seems to state the COC like it works, whereas this reference examined adherence.

L333 - I'd argue that it's testing whale shark responses to tourists etc, rather than revealing implied negative impacts.

L343-345 (and onwards) - Or, potentially, a *changed* management approach, as I've noted in my general comments.

===PREPARING YOUR MANUSCRIPT===

If you have been asked to revise the written English in your submission as a condition of publication, you must do so, and you are expected to provide evidence that you have received language editing support. The journal would prefer that you use a professional language editing service and provide a certificate of editing, but a signed letter from a colleague who is a native speaker of English is acceptable. Note the journal has arranged a number of discounts for authors

using professional language editing services
(<https://royalsociety.org/journals/authors/benefits/language-editing/>).

===PREPARING YOUR REVISION IN SCHOLARONE===

<https://royalsociety.org/journals/authors/author-guidelines/#supplementary-material> to include a suitable title and informative caption. An example of appropriate titling and captioning may be found at https://figshare.com/articles/Table_S2_from_Is_there_a_trade-off_between_peak_performance_and_performance_breadth_across_temperatures_for_aerobic_sc_ope_in_teleost_fishes_/3843624.

Author's Response to Decision Letter for (RSOS-200392.R0)

See Appendix B.

RSOS-200392.R1 (Revision)

Review form: Reviewer 1 (Olivia Lee)

Is the manuscript scientifically sound in its present form?

Yes

Are the interpretations and conclusions justified by the results?

Yes

Is the language acceptable?

Yes

Do you have any ethical concerns with this paper?

No

Have you any concerns about statistical analyses in this paper?

No

Recommendation?

Accept with minor revision (please list in comments)

Comments to the Author(s)

The revised manuscript is much improved particularly in the clarity and readability, and the additions to the discussion with recommendations for follow-up measures that could be taken. I appreciate the time and thoughtfulness put into all of the responses to reviewer comments and hope to see this move forward in the publication process.

Only minor editorial comments below:

Minor editing comments:

Note that order of figures in proof don't line up with order in Figure captions.

Figure 3 - likely doesn't need title caption on graph when explained in the figure caption.

Line 126 page 29 of pdf proof - correct spelling of "sergested" shrimp to "sergestid"

Decision letter (RSOS-200392.R1)

Dear Ms Legaspi

On behalf of the Editors, we are pleased to inform you that your Manuscript RSOS-200392.R1 "In-water observations highlight the effects of provisioning on whale shark behaviour at the world's largest whale shark tourism destination" has been accepted for publication in Royal Society Open Science subject to minor revision in accordance with the referees' reports. Please find the referees' comments along with any feedback from the Editors below my signature.

Please submit your revised manuscript and required files (see below) no later than 7 days from today's (ie 20-Nov-2020) date. Note: the ScholarOne system will 'lock' if submission of the revision is attempted 7 or more days after the deadline. If you do not think you will be able to meet this deadline please contact the editorial office immediately.

on behalf of Prof Pete Smith (Subject Editor)
openscience@royalsociety.org

Associate Editor Comments to Author:

Thank you for submitting your revised paper and for addressing the referees' comments. We've now received one report on your manuscript.

Please ensure that you address these final comments and provide all required files upon re-submission of your paper.

Reviewer comments to Author:

Reviewer: 1

Comments to the Author(s)

The revised manuscript is much improved particularly in the clarity and readability, and the additions to the discussion with recommendations for follow-up measures that could be taken. I appreciate the time and thoughtfulness put into all of the responses to reviewer comments and hope to see this move forward in the publication process.

Only minor editorial comments below:

Minor editing comments:

Note that order of figures in proof don't line up with order in Figure captions.

Figure 3 - likely doesn't need title caption on graph when explained in the figure caption.

Line 126 page 29 of pdf proof - correct spelling of "sergested" shrimp to "sergestid"

===PREPARING YOUR MANUSCRIPT===

===PREPARING YOUR REVISION IN SCHOLARONE===

-- If you have uploaded ESM files, please ensure you follow the guidance at <https://royalsociety.org/journals/authors/author-guidelines/#supplementary-material> to include a suitable title and informative caption. An example of appropriate titling and captioning may be found at [https://figshare.com/articles/Table_S2_from_Is_there_a_trade-off_between_peak_performance_and_performance_breadth_across_temperatures_for_aerobic_sc ope_in_teleost_fishes_/3843624](https://figshare.com/articles/Table_S2_from_Is_there_a_trade-off_between_peak_performance_and_performance_breadth_across_temperatures_for_aerobic_scope_in_teleost_fishes_/3843624).

Author's Response to Decision Letter for (RSOS-200392.R1)

See Appendix C.

Decision letter (RSOS-200392.R2)

Dear Ms Legaspi,

It is a pleasure to accept your manuscript entitled "In-water observations highlight the effects of provisioning on whale shark behaviour at the world's largest whale shark tourism destination" in its current form for publication in Royal Society Open Science.

on behalf of Professor Pete Smith (Subject Editor)
openscience@royalsociety.org

Appendix A

Review comments RSOS-200392

In-water observations highlight the effects of provisioning on whale shark behaviour at the world's largest whale shark tourism destination

General comments:

- Was there a reason why the authors did not use year or some value of seasonality as explanatory variables to predict shark behavior? Alternatively, providing some supporting explanation for pooling observations across years/ seasons would be important.
- Authors could improve clarity on mentioning 'previous visits' as number of days that an individual was seen in the study site.
- Image resolution for Figure 3 is poor in the proof.
- Figure 4 could be moved as electronic supplementary material.
- The section on Tourism Management Implications don't provide any specific guidance or recommendations as a result of the study. Are there any actionable recommendations that the authors can provide?
- In general, the discussion section could use a second round of critical editing for clarity and readability.

Specific comments:

- Line 42: Can authors provide more specific information on economic gains from whale shark tourism to the study area specifically, rather than global economic benefits?
- Line 57-58: What is the difference in the 'displayed avoidance' and 'react (avoidance)' behavior mentioned here?
- Line 77: is it the role of stakeholders to change acceptable standards, or should this statement emphasize the role of local governance/ management agencies? Also, to "delineate limits of accessible change" or is it about delineating limits of acceptable tourism conduct?
- Line 101: Can authors provide more information about whether there is a seasonality in tourism pressure throughout the year? Is there a lower tourist season? If so, what is the change in tourism magnitude?
- Line 129: what is the approximate range of in-water visibility? What months is visibility better?
- Line 134: on section about focal follows authors were not clear about how many volunteers or researchers were involved in collecting data over the study period form 2015-2017. This would give some measure of between-individual data recording potential variability Also, it was not clear if all months from Feb 2015 to May 2017 were included in the research. Ability to break

down differences in study effort – e.g. number of surveys per month/ season over time (e.g. through a table in electronic supplementary material) would be important.

Line 157: Some of the explanatory variables should be more clearly explained. Specifically, on page 9 under Data analysis they describe "current (1-5)" - but do not explain what the values of 1-5 refer to.

Line 158: authors used weekly averages of SST, rather than the daily SST value in the model. Authors could provide a quick description of why weekly average SST values were used rather than the daily values. Also, why the authors chose the 0.5 degree longitude resolution was chosen rather than a higher resolution (e.g. 4 km/ 9 km available from MODIS)

Line 180: Authors should report in results the mean and range of number of whale sharks observed in the study area during each focal follow.

Line 184-185 page 10: Change “ranged from 1 185– 1,321” to “ranged from 1,185 to 1321”

Line 244: Sentence structure. Consider changing to: “Here, we showed the impacts of provisioning to the behaviour of whale sharks in Oslob, and effects of the tourism industry failing to comply with the regulations ; both indicative of strong calls for management intervention”

Line 253: change to “The [majority] of the sharks...”

Line 257: What are significant observations? Was there a statistical significance suggested here?

Line 268: Focus on more active statement, rather than passive e.g. “The GLMM showed that roadblock was a more significant predictor of ...”

Line 271-273: unclear sentence. Should restructure for clarity

Line 275: change to “..associate food from the site and are less likely display avoidance behaviour”

Line 281-283: sentence not clear. Needs to be restructured for clarity

Line 302: change to “... resulting in an overall reduction in the vertical...”

Lines 303-309: Paragraph on explaining whale shark behavior with thermoregulation needs more supporting information and more explanation for surface feeding behavior in relation to thermoregulation; otherwise – I would suggest removing this section from the discussion since the study did not look at movement between depths, or long term migration, or diving behavior specifically.

Lines 321-324: unclear. Sentence needs to be re-written for clarity.

Line 329-331: unclear. Sentence needs to be re-written for clarity.

Figure 2 caption states “frequency distribution of previous visits (days)”; for clarity consider changing to “frequency distribution of previous visits (number of days seen in interaction area)

Appendix B

Large Marine Vertebrates Research Institute Philippines
Cagulada Compound, Brgy. Tejero, Jagna, 6308,
Bohol, Philippines

Dr. Asha de Vos,
Associate Editor,
Royal Society Open Science

October 19th, 2020

Dear Editor and Reviewers,

We would like to thank you for your time and consideration when reviewing this manuscript. We have addressed all of your comments and concerns, and substantially revised the manuscript. Replies to specific comments can be found below.

We hope the changes made have strengthened the manuscript, and that it is now suitable for publication in Royal Society Open Science.

Kind regards,

Christine Legaspi & co-authors

Large Marine Vertebrates Research Institute Philippines
+639950683599 | t.legaspi@lamave.org | www.lamave.org

Review 1 (Anonymous)

Comments to the Author (s):

I think this manuscript provides timely and important information on species behavior and management recommendations for managing the whale shark tourism industry. I think the paper would benefit from a little more introductory explanation about the seasonality of tourism pressure (i.e. is there a low and high tourist season? What is the magnitude of tourism pressure between seasons/ between locations in the Philippines?) I also think the manuscript could use a little more explanation in the data analysis to improve clarity but to also support future meta-analyses of similar studies in the future, and more critical editing of the discussion section to improve clarity in the content.

CL: We thank the Reviewer for their time and consideration. We have revised the manuscript to reflect their concerns. We have highlighted the tourism seasonality in the Introduction and Methods. We are not however assessing the tourism pressure across all sites as this beyond the scope of this paper. We have amended the Discussion for clarity as per the Reviewer's suggestion.

General comments:

-Was there a reason why the authors did not use year or some value of seasonality as explanatory variables to predict shark behavior? Alternatively, providing some supporting explanation for pooling observations across years/ seasons would be important.

CL: We appreciate the Reviewer's comment, and have amended the Methods accordingly to reflect this. We used SST as a seasonal variable, and given its significance in the model outputs, we have discussed its implications in the Discussion.

-Authors could improve clarity on mentioning 'previous visits' as number of days that an individual was seen in the study site.

CL: Agreed and revised. We have clarified and defined 'previous visits' as the number of days that an individual was identified at the site as confirmed through photo-ID in the methods (Data analyses paragraph).

-Image resolution for Figure 3 is poor in the proof.

CL: Higher resolution version submitted with this revision.

-Figure 4 could be moved as electronic supplementary material.

CL: Agreed. Moved as Supplementary Figure 2.

- The section on Tourism Management Implications don't provide any specific guidance or recommendations as a result of the study. Are there any actionable recommendations that the authors can provide?

CL: Agreed. We have altered this section and provided recommendations based on the results.

- In general, the discussion section could use a second round of critical editing for clarity and readability.

CL: This is noted. We have revised the discussion for clarity and readability.

Specific comments:

Line 42: Can authors provide more specific information on economic gains from whale shark tourism to the study area specifically, rather than global economic benefits?

CL: This paragraph is mainly to present the economic benefits of whale shark tourism in general. We have added additional information on the study site to reflect the Reviewer's concerns: "Oslob, Philippines is currently the world's largest non-captive provisioned whale shark dedicated tourism destination, receiving over 500,000 tourists in 2018 with an estimated US\$10 million in ticket sales [15,18, 26]. This community-run site is active year-round, and although some tourist [26] and whale shark [23] seasonality is observed, the sharks are provisioned daily. While the activity offers an avenue for a tourism industry to flourish [24], there is evidence that the current tourism has impacts on the host species [23, 27-28]."

Line 57-58: What is the difference in the 'displayed avoidance' and 'react (avoidance)' behavior mentioned here?

CL: There is no difference. We have revised the sentence to avoid confusion.

Line 77: is it the role of stakeholders to change acceptable standards, or should this statement emphasize the role of local governance/ management agencies? Also, to “delineate limits of accessible change” or is it about delineating limits of acceptable tourism conduct?

CL: The stakeholders that the authors were referring to include the local government, under whose direct management the tourism falls. We have revised the statement for clarity.

Line 101: Can authors provide more information about whether there is a seasonality in tourism pressure throughout the year? Is there a lower tourist season? If so, what is the change in tourism magnitude?

CL: There has not been any assessments on tourism pressure regarding its seasonality. However, we have included some information regarding how the tourism has grown in the description of the study site in the Methods. Information was based from a previous study of Oslob (Ziegler et al, 2019). Based on available data (now Supplementary Figure 1), tourist numbers rapidly grew in a span of 3 years. The graph shows that the numbers vary per month and peaks are different per year. The month with the lowest tourist numbers is September which has been consistent since records are available.

Line 129: what is the approximate range of in-water visibility? What months is visibility better?

CL: We have revised the sentence to avoid confusion with seasonal changes in terms of visibility. The in-water visibility is collected based on the assigned focal follow researcher's estimates. The range was between 2 and 16 m. Visibility can be variable throughout the year but is generally better from April to September.

Line 134: on section about focal follows authors were not clear about how many volunteers or researchers were involved in collecting data over the study period from 2015-2017. This would give some measure of between-individual data recording potential variability. Also, it was not clear if all months from Feb 2015 to May 2017 were included in the research. Ability to breakdown differences in study effort – e.g. number of surveys per month/ season over time (e.g. through a table in electronic supplementary material) would be important.

CL: Effort was not consistent throughout the study period due to staff availability. But, in general, focal follows were done two days a week – on a weekend (busy) and on a weekday (less busy). We have reflected the Reviewer's concern by amending this in the methods: "Effort was not consistent and was dependent on research availability, but weekend (busy) and weekdays were targeted equally to reflect potential differences."

Line 157: Some of the explanatory variables should be more clearly explained. Specifically, on page 9 under Data analysis they describe "current (1-5)" - but do not explain what the values of 1-5 refer to.

CL: We revised the paragraph to address the Reviewer's concern: "Potential exploratory variables included previous visits at the site (the number of days that an individual was identified at the study site, as confirmed through photo-ID), current (1 if there is no current, 2 if there is effort needed to swim, 3 if there is difficulty in swimming, 4 if there is a need to hold on to boats, 5 if it is impossible to swim at all), visibility (m),..."

Line 158: authors used weekly averages of SST, rather than the daily SST value in the model. Authors could provide a quick description of why weekly average SST values were used rather than the daily values. Also, why the authors chose the 0.5 degree longitude resolution was

chosen rather than a higher resolution (e.g. 4 km/ 9 km available from MODIS)

CL: At the time of writing, the resolution available was quite poor for the study site area. This is also reflected by the high cloud cover in the Philippines, and there are consistent gaps in the data. Temperature variation at Oslob is small (3-4 °C) during the year. We therefore used weekly averages to address these gaps.

Line 180: Authors should report in results the mean and range of number of whale sharks observed in the study area during each focal follow.

CL: The authors do not have the number of sharks observed in the study area specifically during the focal follow survey since the assigned researcher only focused on one individual. However we have data of the number of individuals observed during these days. We have added this statement “On the days where focal follow surveys were conducted, there was a mean of 14 ± 5.49 individuals (range 5 – 31 individuals) present in the study area” to add more information.

Line 184-185 page 10: Change “ranged from 1 185– 1,321” to “ranged from 1,185 to 1321”

CL: Agreed and revised.

Line 244: Sentence structure. Consider changing to: “Here, we showed the impacts of provisioning to the behaviour of whale sharks in Oslob, and effects of the tourism industry failing to comply with the regulations ; both indicative of strong calls for management intervention”

CL: Agreed and revised.

Line 253: change to “The [majority] of the sharks...”

CL: Agreed and revised.

Line 257: What are significant observations? Was there a statistical significance suggested here?

CL: Yes. The GLMM results from the previous study (Schleimer et al, 2015) showed that the number of previous visits was a statistically significant predictor of observing vertical feeding.

Line 268: Focus on more active statement, rather than passive e.g. “The GLMM showed that roadblock was a more significant predictor of ...”

CL: Agreed. Revised.

Line 271-273: unclear sentence. Should restructure for clarity

CL: Agreed. Revised.

Line 275: change to “..associate food from the site and are less likely display avoidance behaviour”

CL: Agreed. Revised.

Line 281-283: sentence not clear. Needs to be restructured for clarity

CL: Agreed. Revised.

Line 302: change to “... resulting in an overall reduction in the vertical...”

CL: Agreed. Revised.

Lines 303-309: Paragraph on explaining whale shark behavior with thermoregulation needs more supporting information and more explanation for surface feeding behavior in relation to thermoregulation; otherwise – I would suggest removing this section from the discussion since the study did not look at movement between depths, or long term migration, or diving behavior specifically.

CL: We have revised the paragraph to address the Reviewer’s concern: “When in Oslob, the whale sharks generally spend the majority of their time at the surface because that is where the food is being provisioned [26]. Whale sharks have a need to thermoregulate as ectotherms [24],

and the prolonged exposure to warmer waters [28] particularly during *habagat* when the water is warmer, might also explain their reduced predictability feeding from the food provisioned”.

Lines 321-324: unclear. Sentence needs to be re-written for clarity.

CL: Agreed. Revised.

Line 329-331: unclear. Sentence needs to be re-written for clarity.

CL: Agreed. Revised.

Figure 2 caption states “frequency distribution of previous visits (days)”; for clarity consider changing to “frequency distribution of previous visits (number of days seen in interaction area)

CL: Agreed. Revised.

Reviewer 2 (Simon Pierce)

General comments:

I very much enjoyed reading this study on what has become quite a controversial topic in marine wildlife tourism: provisioning whale sharks. The scientific content has been well implemented. However, I think the discussion, as it is now, reads as more opinionated than the results – taken in isolation – warrant. That said, I am not unsympathetic to the authors’ opinion, and I have a couple of general comments that may be useful to consider.

First, my read of this study – again, taken in isolation – is that the primary takeaway is that ‘habituated whale sharks show minimal avoidance behaviours’. In itself, this doesn’t appear to show a clear negative impact to the sharks. In the introduction (L76-78) the authors contend that local stakeholders should delineate limits of acceptable change – this would be a useful point to expand on within the discussion, in light of the results of this study. A more explicit discussion of real or potential (i.e. based on the broader literature) impacts would also be helpful. This is mentioned in the introduction but could be partly revisited in the discussion based on these new results.

Second, there are clear, systematic breaches of the code of conduct here. In terms of recommendations, I think there is a stark choice for managers – either (a) acknowledge that these ‘rules’ are viewed as unnecessarily precautionary in practice, or (b) take steps to provide an equally systematic solution. To me, the results presented here suggest that a useful measure might be to have all visitors hold onto the boats, or a rope between boats, while feeders slowly move up and down the resulting controlled line of people. Fig 4 shows that RB and S2S cause the most disturbance. Controlling the distance, which appears to be within the feeders control (i.e. L230-232), would reduce most of the ‘events’ listed, while potentially minimising crowding perceptions too, if all the tourists had an unobstructed view of the sharks. That would space out the sharks from one another, and from the people, rather than the free-for-all demonstrated in this study. This is just a suggestion, but, in that spirit, I think more tangible and practical recommendations in the discussion would benefit all parties.

I would also change the MS title to more directly address these points.

I hope this helps.

Simon Pierce.

CL: The authors substantially revised the manuscript and have included your recommendations.

Specific comments:

Abstract:

L15-16 - Relationship *between*. Seasonality rather than periodicity?

CL: Seasonality was not tested as a predictor of a behaviour but rather the number of previous visits. We have redescribed this variable for clarity.

L20-21 - I'd say ‘overcrowding’ rather than ‘high tourism pressure’, as it’s unclear from these results whether this is actually affecting the sharks, but overcrowding is clear based on the site’s own rules.

CL: Agreed, changed to overcrowding.

L23-24 - It doesn't necessarily hold that this is unsustainable, depending on how that term is being defined here. The sharks keep coming back, for one thing, and there isn't necessarily a decline in tourist numbers. You could reword this conclusion to better reflect the direct results of this study.

CL: Agreed, changed to read: "Given the behavioural implications to whale sharks highlighted here and the local community's reliance on the tourism industry, it is imperative to look into improving management strategies to increase tourist compliance and strive for sustainable tourism practices."

Introduction:

L34-35 - This makes their size sound like a bug, rather than a feature – whale sharks are likely to be large to enhance their foraging efficiency.

CL: Agreed, rephrased.

L41 - 'Product' rather than 'market'?

CL: Agreed, rephrased.

L46 - Australia is effectively featured twice in this sentence.

CL: Added Mexico

L50 - Effects rather than implications.

CL: Agreed, changed to effects.

4th Para - How does this study differ and build on previous work, such as refs 21 and 22? Given the amount of work that has taken place at this site, a more detailed introduction would be useful for placing the current study in context.

CL: Revised and added more information regarding the results of previous studies.

Methods:

Table 1 - The new study by Tomita et al. (2020, PLOS ONE) describes this (ER); it's the whole eye, not the pupil.

CL: Revised

Results:

Fig 2 - I don't understand this? Frequency? There are only 46 sharks. Is this a bar for each individual shark? Seems like a different type of graph should be used here.

CL: The figure has been removed to avoid confusion.

Discussion:

The first two sentences are repetitive from the introduction. Better to lead with the main results.

CL: Agreed. Revised.

L273 - Rewording could help make this point clearer.

CL: Agreed. Revised.

L303-304 - I'm not really making the connection between SST as a prediction of occurrence with this particular point. You're seeing more sharks in warmer water; isn't this arguing the opposite, if you're saying the sharks need to cool down following a long exposure to warm-water in the provisioning area?

CL: Revised for clarity: "When in Oslob, the whale sharks generally spend the majority of their time at the surface because that is where the food is being provisioned [26]. Whale sharks have a need to thermoregulate as ectotherms [24], and the prolonged exposure to warmer waters [28] particularly during *habagat* when the water is warmer, might also explain their reduced predictability feeding from the food provisioned."

L329 - Is the Quiros ref relevant here? As used here, it seems to state the COC like it works, whereas this reference examined adherence.

CL: The reference also discussed the function of the code of conduct. This statement is a paraphrased version of a statement in Quiros (2007) discussion: "The Code of Conduct for whale shark interaction is the main line of defence to minimize negative impacts from tourism".

L333 - I'd argue that it's testing whale shark responses to tourists etc, rather than revealing implied negative impacts.

CL: This is noted. We have revised this statement.

L343-345 (and onwards) - Or, potentially , a *changed* management approach, as I've noted in my general comments.

CL: We have revised and added your suggestion.

Appendix C

Large Marine Vertebrates Research Institute Philippines
Cagulada Compound, Brgy. Tejero, Jagna, 6308,
Bohol, Philippines

Dr. Asha de Vos,
Associate Editor,
Royal Society Open Science

November 27th, 2020

Dear Editor and Reviewers,

We would like to thank you for your time when reassessing and accepting this manuscript. We have addressed all of your comments and concerns, and revised the manuscript accordingly. Replies to specific comments can be found below.

We hope you agree with these changes made, and that it is now suitable for publication in Royal Society Open Science.

Kind regards,

Christine Legaspi & co-authors

Large Marine Vertebrates Research Institute Philippines
+639950683599 | t.legaspi@lamave.org | www.lamave.org

Review 1 (Anonymous)

Comments to the Author(s)

The revised manuscript is much improved particularly in the clarity and readability, and the additions to the discussion with recommendations for follow-up measures that could be taken. I appreciate the time and thoughtfulness put into all of the responses to reviewer comments and hope to see this move forward in the publication process.

Only minor editorial comments below:

Minor editing comments:

Note that order of figures in proof don't line up with order in Figure captions.

CL: Agreed. We have updated the figure numbers throughout the manuscript to match their order and captions.

Figure 3 - likely doesn't need title caption on graph when explained in the figure caption.

CL: Agreed. The title on graph has been removed.

Line 126 page 29 of pdf proof - correct spelling of "sergested" shrimp to "sergestid"

CL: Noted and corrected.